# Inhibition of Quorum-Sensing Regulator from *Pseudomonas aeruginosa* Using a Flavone Derivative

**DOI:** 10.3390/molecules27082439

**Published:** 2022-04-10

**Authors:** Yanxuan Xie, Jingxin Chen, Bo Wang, Ai-Yun Peng, Zong-Wan Mao, Wei Xia

**Affiliations:** 1School of Chemistry, Sun Yat-Sen University, Guangzhou 510275, China; xieyx8@mail3.sysu.edu.cn (Y.X.); chenjx256@mail3.sysu.edu.cn (J.C.); ceswb@mail.sysu.edu.cn (B.W.); cespay@mail.sysu.edu.cn (A.-Y.P.); 2MOE Key Laboratory of Bioinorganic and Synthetic Chemistry, Sun Yat-Sen University, Guangzhou 510275, China

**Keywords:** *Pseudomonas aeruginosa*, quorum sensing, *las* system, flavone derivative, inhibitor

## Abstract

Quorum sensing (QS) is a cell-to-cell communication process that controls bacterial collective behaviors. The QS network regulates and coordinates bacterial virulence factor expression, antibiotic resistance and biofilm formation. Therefore, inhibition of the QS system is an effective strategy to suppress the bacterial virulence. Herein, we identify a phosphate ester derivative of chrysin as a potent QS inhibitor of the human pathogen *Pseudomonas aeruginosa* (*P. aeruginosa*) using a designed luciferase reporter assay. In vitro biochemical analysis shows that the chrysin derivative binds to the bacterial QS regulator LasR and abrogates its DNA-binding capability. In particular, the derivative exhibits higher anti-virulence activity compared to the parent molecule. All the results reveal the potential application of flavone derivative as an anti-virulence compound to combat the infectious diseases caused by *P. aeruginosa*.

## 1. Introduction

Quorum sensing (QS) is a cell-to-cell signaling mechanism that involves the production, release and response to auto-inducers (AIs) signaling molecules. As the bacterial cell density increases, the AIs accumulate in the surrounding environment. Bacterial cells monitor the concentration changes of AIs to track changes in cell numbers and, thereby, alter corresponding gene expression [1]. QS has been known to control a series of bacterial processes that are important for bacterial survival and infection, including bioluminescence, sporulation, antibiotic resistance, biofilm formation and virulence factor production [2,3,4]. Therefore, the QS system is an attractive target for drug development to treat bacterial infectious diseases [5,6].

*Pseudomonas aeruginosa* is one of the common human opportunistic pathogens that causes a wide spectrum of infectious diseases, such as urinary, burn and respiratory infections [7,8]. *P. aeruginosa* displays intrinsic resistance to many antibiotics and could also acquire actively genetic mutations for further resistance [9]. Therefore, *P. aeruginosa* usually develops resistance to most conventional antibiotics [10]. *P. aeruginosa* has a complex QS network that consists of four connected signaling systems: *las*, *rhl*, *pqs* and *iqs* [11,12]. The *las* and *rhl* systems are two well-characterized QS systems. The two systems use N-(3-oxododecanoyl) homoserine lactone (3O-C12HSL) and N-butyryl homoserine lactone (C4-HSL) as their AI molecules respectively, with LasR and RhlR proteins as their respective receptors. The third system, *pqs*, produces 2-heptyl-3-hydroxy-4-quinolone (PQS) AI and uses PqsR as the cognate receptor. The fourth system is *iqs*, which produces 2-(2-hydroxyphenyl)-thiazole-4-carbaldehyde (IQS) as AIs. However, the regulatory mechanisms of the *iqs* system are less well understood. These QS systems are organized in a hierarchical manner, with *las* governing the expression of the *rhl* and *pqs* systems. Therefore, the *las* system is considered to be at the top of the hierarchy. Such a complex hierarchical network may enable *P. aeruginosa* to adapt to various environmental signals and biological stresses [13]. Given its essential role in bacterial QS network, the *las* system has become a potential target for anti-microbial therapy [5,14].

Flavonoids are a group of natural products that exhibit a wide range of biological activities, such as anti-inflammatory, anti-microbial and anti-oxidant activities [15,16,17]. A recent study showed that multiple flavonoids could interfere with the *las* system of *P. aeruginosa* through binding to the LasR regulator, implying the potential to use flavonoids as a QS inhibitor [18]. However, higher concentrations of flavonoids are usually required for QS inhibition. For example, the required inhibitory concentrations for mosloflavone and baicalin are both higher than 300 μM, hindering their practical applications [19]. In this study, we screened a panel of flavonoids and xanthone derivatives using a self-constructed *E. coli* luciferase reporter strain and identified a chrysin derivative as a potent inhibitor of the *P. aeruginosa las* QS system. The results of in vitro and in vivo experiments demonstrated that the inhibitor allosterically inhibits the binding of the LasR regulator to its cognate DNA promoter. In particular, the identified derivative exhibits much higher activity in the suppression of bacterial virulence, biofilm formation and motility when compared to the previously reported flavonoids, indicative of potential application in treatment of *P. aeruginosa* infection.

## 2. Results

### 2.1. Screening for QS Inhibitory Compounds Using an Escherichia coli Reporter Strain

To screen for the potential inhibitors for *P. aeruginosa las* QS system, we first constructed an *Escherichia coli* reporter strain. The *E. coli* strain contains arabinose-inducible LasR and a *lasB* promoter fused to the luciferase gene cassette (*lasB-luxABCDE*). Therefore, the expression of the luciferase gene cassette could be controlled by LasR (Figure 1a). The *E. coli* reporter strain could produce a high level of bioluminescence only in the presence of arabinose and the AI molecule 3O-C12HSL (Appendix A).

To identify potential LasR inhibitors, we first screened a self-synthesized 48-molecule library containing different flavone and xanthone derivatives. It has been reported that xanthones exhibited good anti-cancer, anti-bacterial and quorum-sensing-inhibitory activities [20,21,22]. Since xanthone and flavonoid both contained the 4H-chromen-4-one moiety that is an important pharmacophore in medicinal chemistry [23,24], xanthones derivatives were also included in the screening (Appendix A). The molecules that could significantly decrease the bioluminescence emission of the reporter strain were chosen. In total, A12, C5, D1, D4, D5, D6, D7, E5, E6 and E7 were selected and then applied for a second round of screening using an *E. coli* control strain that contained the luciferase gene cassette fused to a constitutive *tac* promoter (*tac-luxABCDE*). The molecules that interfered with the luciferase activity could be excluded (Figure 1b). After two rounds of screening, the compounds D5, D6, D7, E5, E6 and E7 could decrease the bioluminescence emission of reporter strain without affecting the signals in the control strain (Appendix A). Therefore, three xanthone derivatives (D5, D6, D7) and three chrysin derivatives (E5, E6, E7) were obtained for further analysis (Figure 1c).

We further verified the suppression effect of those identified compounds. Indeed, all these six compounds could substantially reduce the bioluminescence signals of the *E. coli* reporter strain in a concentration-dependent manner (25 to 100 μM) (Appendix A). However, all these six compounds also interfered with the bioluminescence of the *E. coli* control strain to some extent (Appendix A). Therefore, we further tested whether the six compounds could interfere with the LasR regulator in vitro and in vivo experiment.

### 2.2. Interaction between Identified Compound and Transcriptional Regulator LasR

We subsequently investigated whether these identified compounds could bind to the LasR regulator in vitro using the fluorescent thermal shift assay (FTSA) as described previously [25]. To this end, we first overexpressed and purified full-length LasR and the ligand binding domain (LBD) of LasR. Since LasR and related proteins are not folded and are insoluble without the cognate AI molecule 3O-C12HSL, the two proteins were overexpressed in the presence of 3O-C12HSL and purified as the 3O-C12HSL-bound form [26]. For FTSA analysis, 10 μM purified LasR–LBD was incubated with 100 μM 3O-C12HSL in the presence or absence of 200 μM compounds at 4 °C for 30 min. The melting curves of proteins were subsequently recorded on a StepOnePlus thermocycler as described previously [16]. The fluorescent intensity was monitored as the samples were heated from 25 to 99 °C at an increment of 1 °C/min (Appendix A). As shown in Table 1, all these six identified compounds caused the rise of the melting temperature of LasR–LBD with ∆T values of 0.14 to 1.43 °C. In particular, the incubation of compound E6 led to the most notable change of melting temperature with a ∆T value of 1.43 °C. It is worth noting that these identified compounds could enhance the thermostability of LasR–LBD even in the presence of an excess of 3O-C12HSL molecules, implying that these compounds might not compete for the AI binding site.

Since LasR could bind to its cognate DNA promoter to regulate the QS system, we subsequently investigated whether the identified compounds could interfere with the DNA-binding capability of LasR using the electrophoretic mobility shift assay (EMSA) [27]. The *lasB* promoter (P*_lasB_*) was used as a DNA probe and amplified using the *P. aeruginosa* genomic DNA as a template. In general, 30 ng P*_lasB_* DNA or negative control (*hemO* gene promoter P*_hemO_*) were incubated with gradient concentrations of full-length LasR (0 to 1000 nM). Indeed, a significant shift of P*_lasB_* DNA was observed after incubation of LasR. Specifically, 1 μM LasR was sufficient to bind all the P*_lasB_* probe in EMSA (Appendix A). In contrast, no perturbation was observed for the negative control P*_hemO_* probe (Appendix A).

For the compound perturbation assay, 1 μM LasR was incubated with 30 ng P*_lasB_* DNA in the presence of 100 μM identified compounds or a well-characterized QS inhibitor (*Z*-)-4-bromo-5-(bromomethylene)-2(5H)-furanone (furanone C-30) [28]. The mixture was then analyzed by EMSA. As shown in Figure 2, the binding between LasR and P*_lasB_* DNA was completely abrogated in the presence of furanone C-30. Incubation of the identified compounds caused partial dissociation between the LasR and DNA probe. Typically, D5, D6, E6 and E7 are the most effective compounds to disrupt the LasR–DNA complex, whereas D7 and E5 are less potent. Collectively, all these in vitro biochemical results demonstrated that the identified compounds could directly bind to LasR and prevent its binding to the *lasB* promoter DNA.

### 2.3. Identified Compounds Inhibit Biofilm Formation and Virulence Factor

Since LasR is the crucial regulator of the biofilm formation and virulence factor in *P. aeruginosa*, we sought to investigate whether the six compounds could interfere with the bacterial biofilm formation. The biofilm inhibition assays were performed using the 96-well polystyrene microtiter plate assay as described previously [29]. Compound E5, E6 and E7 were the derivatives of chrysin, and it has been reported that chrysin could substantially inhibit the biofilm formation of *P. aeruginosa* [30]. Therefore, chrysin was used as a positive control in the biofilm and virulence inhibition assay. In brief, *P. aeruginosa* was cultured in the presence of 50 μM chrysin or identified compounds for 24 h. After incubation, planktonic bacterial cells were removed by washing with PBS, and the adherent biofilms were stained with crystal violet. Then, the stained biofilms were solubilized with acetic acid and quantified. As shown in Figure 3a, all the six compounds exhibited a similar inhibitory effect on the biofilm formation of *P. aeruginosa* that was comparable to that of chrysin.

Furthermore, the effects of the identified compounds on the virulence factor expression of *P. aeruginosa* were also examined. The production of three major *P. aeruginosa* virulence factors that are under the control of the bacterial QS systems was measured, including the amounts of LasA protease, pyocyanin and LasB elastase. In brief, an overnight *P. aeruginosa* culture was diluted with fresh LB medium to a final OD_600_ value of 0.05. Then, chrysin or identified compounds were added into the culture to a final concentration of 100 μM. The bacteria were further cultured for 24 h before the supernatant was collected by centrifugation. The secreted bacterial pigment pyocyanin in the supernatant was extracted and measured as described previously [31,32]. The secretion of LasA protease and LasB elastase was measured using the azocasein assay and the elastin-Congo red assay, respectively [33,34]. As shown in Figure 3b, E6 is the most potent compound to suppress the production of three QS-controlled virulence factors in the bacterial supernatant. Typically, the secreted LasA protease, pyocyanin and LasB elastase were decreased by 18.8%, 63.8% and 42.2% after incubation with the E6 compound. All these results revealed that compound E6 is the most potent inhibitor of the *P. aeruginosa las* system. The compound binds to LasR regulator and abrogates the interaction between LasR and P*_lasB_* DNA, resulting in suppression of the virulence factor expression and biofilm formation of *P. aeruginosa*.

### 2.4. Characterization of the Inhibitory Activity of Compound E6

Therefore, we subsequently focused on the E6 compound and characterized its anti-QS capability. Further analysis verified that the E6 compound suppressed the bacterial virulence factor production in a concentration-dependent manner (Figure 4a and Appendix A). The E6 compound has very low toxicity toward both bacterial and mammalian cells. Even 100 μM of the E6 compound has no effect on the growth of *P. aeruginosa*, indicating that the E6 compound is a typical anti-virulence inhibitor and has no bactericidal or bacteriostatic effect (Appendix A). Moreover, the IC_50_ values of compound E6 against human hepatocyte LO2 cells, lung epithelial BEAS-2B cells, lung fibroblast WI-38 cells and lung carcinoma A549 cells were all around 200 μM, indicative of low cytotoxicity of the compound (Appendix A).

Since the E6 compound could substantially attenuated the virulence factor expression of *P. aeruginosa*, we subsequently investigated whether the transcriptional levels of the virulence-related genes are perturbed by the compounds. The transcriptional levels of eight genes were examined by quantitative PCR (qPCR), including three QS transcriptional regulators (*lasR*, *rhlR* and *pqsR*), two secreted proteases (*lasA* and *lasB*) and three enzymes (*rhlI*, *phzM* and *phzS*) that are involved in AI and pyocyanin biosynthesis [32,35]. Consistent with the virulence assay, the qPCR results indicated that the transcriptional levels of these virulence-related genes were significantly attenuated after E6 compound treatment (Figure 4b).

As the motility of *P. aeruginosa* is positively regulated by the *las* and *rhl* QS signals [36], we applied a plate-based assay to assess the effect of E6 compound on the motility of *P. aeruginosa* as described previously [37]. In brief, *P. aeruginosa* was cultured overnight in LB medium supplemented with 50 μM of compound E6. The next day, bacteria were inoculated onto LB agar plates with or without 50 μM of compound E6. The diameters of the swarming and swimming motility zones were measured after further incubation at 37 °C for the indicated time. As shown in Figure 4c, the diameters of bacterial swimming and swarming motility zones were 2.07 ± 0.06 and 2.33 ± 0.12 cm, respectively in the control group. In contrast, the bacterial motility was significantly retarded after incubation with compound E6, with swimming and swarming motility zone diameters decreased to 1.47 ± 0.12 and 1.60 ± 0.10 cm. The results demonstrated that compound E6 could also suppress the QS-regulated bacterial swimming and swarming phenotype.

Finally, rigid-body docking of compound E6 to the LasR–LBD structural model revealed that compound E6 could probably bind to the connecting region of the LasR–LBD dimer. Typically, the flavonoid forms hydrophobic interaction with surrounding residues Leu125 and Gly123 from the adjacent protomer. Hydrogen bonds are also formed between the backbone amide NH of the Asp43 and Ser44 and the carbonyl group of E6. In particular, the phosphoramidate group forms hydrogen bonds with the sidechains of Tyr47 and Thr80 (Figure 4d). Intriguingly, compound E6 does not occupy the ligand site where the AI molecule binds, indicative of an allosteric inhibitory mechanism against LasR functions.

## 3. Discussion

Anti-microbial resistance (AMR) has become a huge threat to global public health. Great efforts have been devoted to the development of new anti-microbial reagents. Accumulating evidence shows that anti-virulence therapy represents an appealing approach to combat AMR. Distinct from the conventional antibiotics that inhibit bacterial essential cellular functions, anti-virulence therapy aims to attenuate the bacterial virulence rather than to kill bacteria directly, which presumably reduces the selective pressure for the emergence of drug resistance. The bacterial quorum-sensing (QS) system exerts an important role in virulence expression, antibiotic resistance and biofilm formation in bacteria. In particular, QS blockade abrogates the virulence without affecting bacterial growth. Therefore, QS represents an attractive target for anti-virulence therapy.

For the discovery of QS inhibitors, activity-based screening is the most widely applied approach by monitoring the change of QS-based phenotypes, such as virulence factor secretion and biofilm formation. However, the QS regulatory network in *P. aeruginosa* is organized in a multi-layered hierarchy containing four different interconnected QS systems, including *las*, *rhl*, *pqs* and *iqs*, and it is difficult to identify specific inhibitors for each QS system using the activity-based screening approach. Therefore, the construction of individual QS regulatory circuit of *P. aeruginosa* in *E. coli* could facilitate the discovery of QS inhibitors. The *las* reporter system described in this study could be readily extended to other QS systems in *P. aeruginosa*.

Previous studies demonstrated that certain natural products, such as flavonoids, exhibit a wide range of anti-microbial activity. However, relatively high concentrations of flavonoids are required for bacterial inhibition. For example, catechin, a flavonoid from *Combretum albiflorum*, can reduce the production of pyocyanin in *P. aeruginosa* by 50% at 250 μM [38]. Quercetin can cause more than 95% inhibition of biofilm formation in *P. aeruginosa* at 281 μM but hardly inhibits the production of pyocyanin [39]. A study by Luo et al. indicated that 573 μM baicalin significantly attenuates the secretion of LasA protease and LasB elastase by 80% and 90% [16]. A recent study also demonstrated that calycopterin from *Marcetia latifolia* inhibits swarming motility in *P. aeruginosa* by 35% at 140 μM [40]. However, the required high concentration prevents the practical application of these flavonoids. In this study, we identified six compounds that could attenuate the bioluminescence of the reporter stain. Particularly, 50 μM compound E6 was sufficient to attenuate the virulence factor expression (LasA protease, pyocyanin and LasB elastase) (Figure 5), biofilm formation and motility (swimming and swarming) of *P. aeruginosa.* The concentration required was much lower than that of previous flavonoids. The results indicated that the derivative possess higher anti-virulence activity than the previously reported flavonoids. Molecular docking results revealed that the phosphoramidate group of compound E6 could probably bind to the residue Tyr47, which is located in the loop L3 of LasR and covers the LasR ligand binding pocket [41]. The docking results implied that compound E6 binding may perturb the conformation of Tyr47 and destabilized the ligand-bound LasR dimer, thereby inducing the dissociation of LasR from bound DNA.

In summary, the phosphate ester derivative of chrysin could be a novel anti-QS compound against *P. aeruginosa* infection. Our findings also showed that the proper chemical modification of flavonoids could substantially enhance their anti-microbial activity and show potential application in anti-virulence therapy.

## 4. Materials and Methods

### 4.1. E. coli Reporter Strain Construction

To construct the LasR expression plasmid, the *P. aeruginosa lasR* gene was amplified by PCR using *P. aeruginosa* PAO1 genomic DNA as a template. The PCR product was digested with *EcoRI* and *XbaI* restriction enzymes and ligated into the pBAD30 vector with an arabinose-inducible promoter. To construct the luciferase reporter plasmid, the *luxABCDE* operon from *Photorhabdus luminescens* was amplified by PCR using the pBAV1k vector as a template [42]. Then the *lasB* promoter and *luxABCDE* operon were fused by bridging PCR and sub-cloned into the pET28a (Novagen) vector. The two plasmids were then co-transformed into DH5α *E. coli* cells to construct the *E. coli* reporter strain.

To construct the *E. coli* control strain that constitutively expresses luciferase, the *tac* promoter was fused to the upstream of the *luxABCDE* operon by bridging PCR. Then, the PCR product was cloned into the pET28a vector. Then the plasmid was transformed into DH5α *E. coli* cells. All the primers used for PCR amplification are listed in Appendix A.

### 4.2. Screening for Inhibitors

The *E. coli* reporter strain was grown overnight in 5 mL LB medium containing 30 μg/mL kanamycin and 100 μg/mL ampicillin at 37 °C. Overnight culture was diluted 1:100 in 20 mL of fresh LB medium supplemented with proper antibiotics and grown at 37 °C until OD_600_ reached 0.5. L-arabinose (Sigma-Aldrich, St. Louis, MO, USA) and 3O-C12HSL (Sigma-Aldrich) were added into bacterial culture to a final concentration of 0.1% (*w*/*v*) and 10 μM, respectively. Subsequently, 150 μL of the reporter strain culture was added to each well of a 96-well white plate (Corning). Next, 2.5 mM stock compound dissolved in DMSO was added to the well with a final concentration of 100 μM. The compounds used for screening were obtained using an in-house compound library [43,44,45,46,47]. Plates were incubated at 30 °C for another 4 h with shaking. The bioluminescence and the absorption at 600 nm (OD_600_) were recorded on a Cytation 3 cell imaging multi-mode plate reader (BioTek).

### 4.3. LasR and LasR LBD Protein Expression and Purification

The full-length LasR and the ligand binding domain (LBD) of LasR (1 to 173) were PCR-amplified using *P. aeruginosa* PAO1 genomic DNA as a template. Then two PCR products were digested by *EcoRI* and *HindIII* restriction enzymes and then ligated into pET47b (+) (Novagen) vector. The constructed plasmids were transformed into BL21 *E. coli* cells and cultured in LB medium supplemented with 30 mg/L kanamycin, 10 μM 3O-C12HSL and 50 mM 3-(N-morpholino) propanesulfonic acid (MOPS), pH 7.0. Then ispropyl β-D-1-thiogalactopyranoside (IPTG) was added into the culture to a final concentration of 0.1 mM to induce the proteins’ expression. The bacteria were further cultured at 18 °C overnight. Cells were pelleted by centrifugation at 4000× *g* for 15 min and resuspended in lysis buffer (20 mM Tris-HCl, 500 mM NaCl, 1 mM β-mercaptoethanol, 1 mM phenylmethylsulfonyl fluoride, pH 8.0). The cell pellets were lysed by sonication, and the supernatant was isolated by centrifugation at 20,000× *g* for 20 min at 4 °C. The supernatant was loaded onto a Ni-NTA affinity column (Qiagen) pre-equilibrated with buffer A (20 mM Tris-HCl, 500 mM NaCl, 50 mM imidazole, pH 8.0), and the protein was eluted with buffer B (20 mM Tris-HCl, 500 mM NaCl, 500 mM imidazole, pH 8.0). The eluted fractions were dialyzed against buffer C (20 mM Tris-HCl, 250 mM NaCl, 1 mM β-mercaptoethanol, pH 8.0) overnight in the presence of PreScission protease (molar ration of protease to target protein is 1:100) to remove the His-tag. The digested protein was pooled and loaded onto a size-exclusion chromatography Superdex S200 column (GE Healthcare) pre-equilibrated with buffer D (20 mM Tris-HCl, 250 mM NaCl, 1 mM DTT, pH 8.0). The final eluted protein was pooled, concentrated to 2 mg/mL, flash frozen in liquid N_2_ and stored at −80 °C. All the primers used in PCR amplification are listed in Appendix A.

### 4.4. Thermal Shift Assay

The thermal shift assay samples were prepared in a reaction buffer (20 mM Tris-HCl, 250 mM NaCl, 1 mM DTT, pH 8.0). The 20 μL reactions samples contained 10 μM LasR–LBD protein, 100 μM 3O-C12HSL, 200 μM compounds and 1:125 (*v*/*v*) diluted SYPRO Orange fluorescent dye (Sigma-Aldrich). The mixtures were added in an 8-tube strip (Applied Biosystems) and incubated at 4 °C for 30 min. Then the melting curves of the samples were recorded on a StepOnePlus Real-Time PCR instrument (Life Technologies). Samples were heated from 25 to 99 °C at an increment of 1 °C/min. The melting temperatures of the protein samples were analyzed by the Protein Thermal Shift software.

### 4.5. Electrophoretic Mobility Shift Assay

The *P. aeruginosa lasB* promoter DNA fragment (P*_lasB_* contains 286 bp) was amplified by PCR using *P. aeruginosa* PAO1 genomic DNA as a template. For LasR–P*_lasB_* binding analysis, 30 ng purified P*_lasB_* was incubated with gradient concentrations of purified full-length LasR (50–1000 nM) in binding buffer (20 mM Tris-HCl, 300 mM NaCl, 5 mM MgCl_2_, 10 μM 3O-C12HSL, 5% glycerol, pH 8.0). For the chemical compound perturbation assay, 30 ng purified P*_lasB_* was incubated with 1 μM of full-length LasR in binding buffer. Then identified compound was added into each EMSA sample with a final concentration of 100 μM and incubated for an additional 30 min at 4 °C. The EMSA samples were then subjected to electrophoresis on 6% native polyacrylamide gels at 120 V for 95 min. After electrophoresis, gels were stained by GeneFinder nucleic acid staining solution (Xiamen Zhishan) for 20 min in the dark and visualized using a blue-light trans-illuminator (Syngene).

### 4.6. P. aeruginosa Biofilm Inhibition Assay

For the biofilm inhibition assay, an overnight *P. aeruginosa* bacterial culture was inoculated in fresh LB medium supplemented with 0.5% (*w*/*v*) glucose and grown at 37 °C until OD_600_ reached 0.05. Then 100 μL cell cultures were transferred into each well of the 96-well U-bottom microtiter plate (NEST) with or without 50 μM identified compounds. The 96-well plates were further incubated at 37 °C for 24 h. The next day, non-adherent cells were removed from the plate, and the plate was gently washed with sterile phosphate-buffered saline (PBS) three times. The remaining adherent biofilms were dried and further stained with 0.1% (*w*/*v*) crystal violet solution for 20 min. The excess dye was removed by rinsing the plate three times with PBS. Stained biofilms were then solubilized by adding 33% acetic acid and quantified by measuring the absorbance at 595 nm. Each experiment was repeated four times.

### 4.7. Measurement of P. aeruginosa’s Secreted Virulence Factors

*P. aeruginosa* bacteria were cultured in fresh LB medium until OD_600_ reached 0.05. Then, identified compounds were added into bacterial culture to a final concentration of 100 μM. The bacterial culture was further incubated at 37 °C with shaking for 24 h, the supernatant was collected by centrifugation at 4000× *g* for 10 min at 4 °C and filtered by a 0.22 μm syringe filter.

For the LasA protease assay, 5 μL of filtered supernatant was mixed with 1 mL reaction buffer (50 mM Tris-HCl, 0.5 mM CaCl_2_, pH 7.5) supplemented with 0.5% (*w*/*v*) azocasein. The mixture was incubated at 37 °C for 15 min. Subsequently, 500 μL of 10% trichloroacetic acid was added and incubated for 30 min at room temperature to stop the proteolytic reaction. Following this, the supernatant was collected by centrifugation at 10,000× *g* for 20 min and the absorbance at 440 nm was measured on a Cytation 3 cell imaging multi-mode plate reader.

The amount of secreted LasB elastase in the supernatant was measured by the elastin-Congo red assay. In brief, 700 μL supernatant was incubated with 700 μL elastin-Congo red solution (10 mg/mL Congo red elastase substrate in 100 mM Tris-HCl, 1 mM CaCl_2_, pH 7.5) at 37 °C for 3 h with shaking. The supernatant was then collected by centrifugation at 16,000× *g* for 10 min. The amount of secreted elastase was quantified by measuring the absorbance at 495 nm.

To determine the amount of secreted pyocyanin in the bacterial supernatant, supernatant (1 mL) was mixed with an equal volume of chloroform three times to extract pyocyanin. The chloroform layer was collected and then mixed with 300 μL HCl (0.2 M). After shaking, the water phase was collected and transferred into 96-well plates. The UV absorbance of the samples at 520 nm were measured on a Cytation 3 cell multi-mode plate reader.

### 4.8. P. aeruginosa Motility Assay

*P. aeruginosa* bacteria were cultured overnight in LB medium supplemented with or without 50 μM of compound E6. To monitor the bacterial swimming motility, a sterile pipette was dipped in the overnight bacterial culture and stabbed into a swimming LB agar plate (0.2% (*w*/*v*) casein hydrolysate and 0.3% (*w*/*v*) agar) supplemented with or without 50 μM of compound E6. Then the plates were incubated at 37 °C for 10 h, and the swimming motility zone was measured. 

Swarming motility was measured by inoculating the overnight culture onto a swarming LB agar plate (0.2% (*w*/*v*) casein hydrolysate, 0.5% (*w*/*v*) glucose and 0.5% (*w*/*v*) agar) supplemented with or without 50 μM of compound E6. Then the plates were incubated at 37 °C for 18 h, and the swarming motility zone was measured.

### 4.9. Measurement of Virulence-Related Gene Transcription by Quantitative Real-Time PCR

*P. aeruginosa* was grown to log phase and diluted in fresh LB medium to a final OD_600_ of 0.1. Then, compound E6 was added into the culture to a final concentration of 50 μM. After further incubation for 4 h, the total RNA was extracted from *P. aeruginosa* cells using the SV total RNA extraction kit (Promega) according to the manufacturer’s instructions. The integrity of the RNA was assessed by 1% agarose gel electrophoresis, and the concentration of RNA was quantified by NanoDrop 2000 (Thermo Scientific, Waltham, MA, USA). Next, the cDNA was synthesized by reverse transcription using the GoScript reverse transcriptase system (Promega) following the manufacturer’s instructions. The transcription level of the detected gene was subsequently determined by real-time PCR using GoTaq qPCR Master Mix kit (Promega) on a StepOnePlus Real-Time PCR system (Life Technologies, Carlsbad, CA, USA). The rrsA (16S rRNA) was used as an internal control. All the experiments were conducted in triplicate, and relative expression levels were measured using the 2^−ΔΔCt^ method. All the primers used in qPCR are listed in Appendix A.

### 4.10. Effect of the Identified Compound E6 on Bacterial Growth

*P. aeruginosa* growth curves were measured by a BioTek Cytation 3 plate reader using the dynamics model. Briefly, a single colony of *P. aeruginosa* PAO1 strain was picked from the LB agar plate and cultured in LB medium. After OD_600_ of the culture reached 1 (~1.5 × 10^8^ CFU/mL), the culture was diluted 10 times to an OD_600_ value of 0.1 with fresh LB medium with different concentrations of compound E6 (50 μM, 100 μM). The bacterial cultures were then incubated at 37 °C in a plate reader with orbital shaking at 220 rpm in a 24-well plate. The bacterial growth was monitored via OD_600_ values at 60 min intervals.

### 4.11. Mammalian Cell Cytotoxicity of Compound E6

The cell cytotoxicity of compound E6 toward mammalian cells was measured by 3-(4,5-dimethylthiazol-2-yl)-2,5-diphenyltetrazolium bromide (MTT) assay. In brief, human hepatocyte LO2 cells, lung epithelial BEAS-2B cells, lung fibroblast WI-38 cells and lung carcinoma A549 cells were grown in Dulbecco’s modified Eagle’s medium (DMEM, Gibco) containing 10% FBS and 100 units/mL penicillin–streptomycin. Cells were seeded into 96-well plates (10^4^ cells per well) and cultured at 37 °C for 18 h. Then, the cells were incubated with compound E6 in a gradient concentration (0.78125 to 200 μM) at 37 °C for additional 44 h. Subsequently, the cell culture media was removed, and the plates were washed three times with PBS. Next, 90 μL of fresh DMEM media and 10 μL of 5 mg/mL MTT solution was added into each well and incubated at 37 °C for 4 h. Subsequently, the solution was removed, and 100 μL DMSO was added into each well and the absorbance at 490 nm was measured. Each experiment was performed in triplicate.

### 4.12. Molecular Modeling

For molecular docking, the structure model of *P. aeruginosa* LasR–LBD was downloaded from the RCSB PDB database (PDB ID 3IX3). The structure of E6 was obtained using ChemDraw 3D software. The structure files of the protein and ligand were converted to pdbqt files by MGLTool software (The Scripps Research Institute, La Jolla, CA, USA). The docking experiments were performed with a grid box covering LasR–LBD monomer structure and the connecting region of the LasR–LBD dimer by Autodock vina software [48]. The docking model with the lowest energy was selected, and the structure figures were generated using PyMol software.

## Figures and Tables

**Figure 1 molecules-27-02439-f001:**
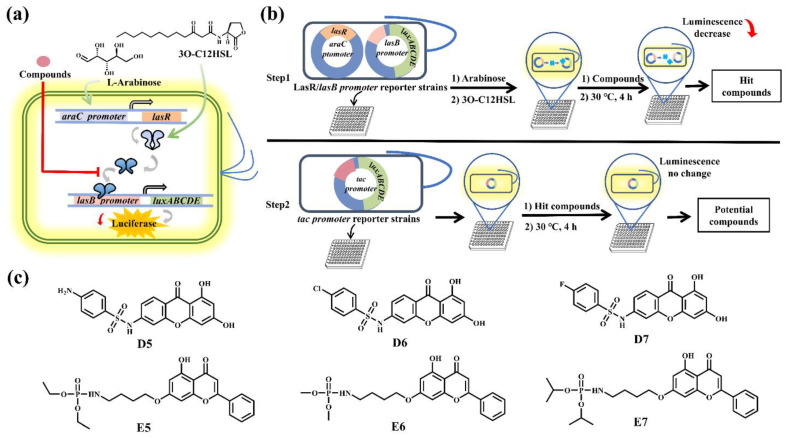
Schematic illustration of the *E. coli* luciferase-based screening assay. (**a**) The construction of the *Escherichia coli* luciferase reporter strain. (**b**) The procedure of screening of potential inhibitors toward the *P. aeruginosa las* system. (**c**) Chemical structures of the identified six compounds that attenuate the bioluminescence of *E. coli* reporter strain, including three xanthone derivatives (D5, D6, D7) and three flavone derivatives (E5, E6, E7).

**Figure 2 molecules-27-02439-f002:**
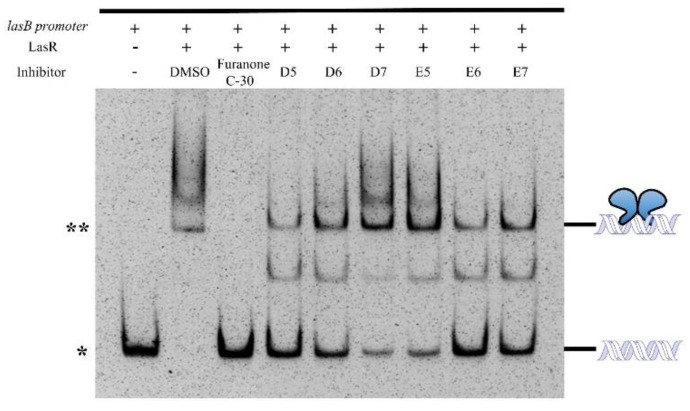
Electrophoretic mobility shift assay of full-length LasR protein binding to *lasB* promoter DNA. Furanone C-30 is a synthetic furanone bacterial quorum-sensing inhibitor [28]. * Indicates the free DNA band, ** indicates the DNA–LasR complex band.

**Figure 3 molecules-27-02439-f003:**
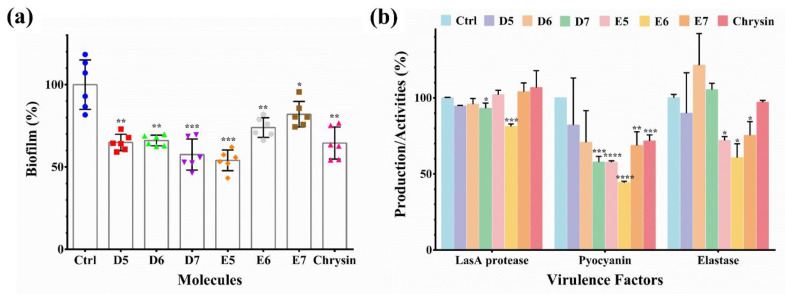
(**a**) Inhibition of *P. aeruginosa* biofilm formation by different compounds. The biofilm content of the control group was set as 100%. The biofilm content in other groups was normalized to the control. (**b**) Inhibition of *P. aeruginosa* virulence factor by different compounds. All experiments were performed in triplicate. Results are shown as mean ± sd. * *p* < 0.05, ** *p* < 0.01, *** *p* < 0.001, **** *p* < 0.0001.

**Figure 4 molecules-27-02439-f004:**
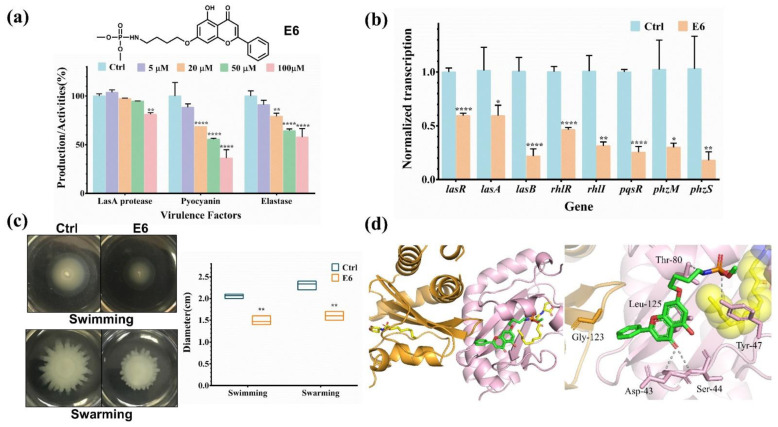
(**a**) Inhibition of *P. aeruginosa* virulence factor expression by increasing gradient concentrations of compound E6 (5 to 100 μM). (**b**) Quantitative transcript analysis of QS-regulated genes treated with or without 50 μM compound E6. (**c**) The swimming and swarming motility assays of *P. aeruginosa* with and without 50 μM of compound E6. (**d**) Docking model of compound E6 binding to LasR–LBD; hydrogen bonds are indicated by the gray dashed line. The molecular docking model indicates the plausible allosteric inhibitory mechanism. Results are shown as mean ± sd. * *p* < 0.05, ** *p* < 0.01, **** *p* < 0.0001.

**Figure 5 molecules-27-02439-f005:**
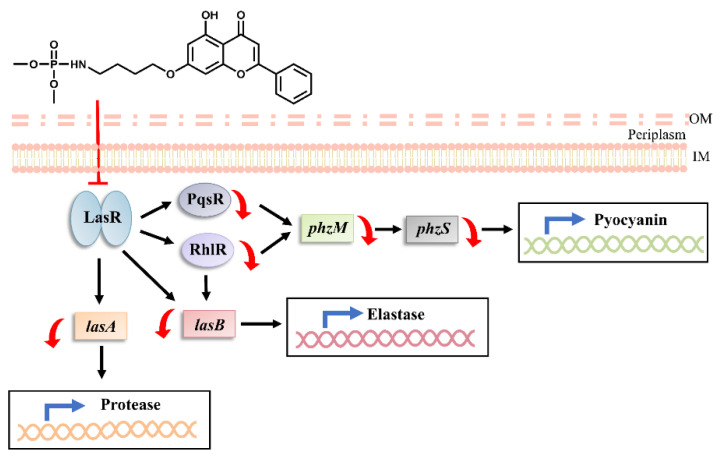
Inhibitory mechanism of flavone derivative toward *P. aeruginosa las* quorum-sensing system. The scheme shows the inhibition of different regulatory pathways for *P. aeruginosa* virulence expression.

**Table 1 molecules-27-02439-t001:** Identified compounds that bind to the LasR–LBD.

Protein and Molecule	Melting Temperature (°C)	ΔTm (°C)
LasR–LBD	59.18	
LasR–LBD + DMSO	59.17	0.00
LasR–LBD + D5	60.46	1.28
LasR–LBD + D6	60.09	0.92
LasR–LBD + D7	59.73	0.55
LasR–LBD + E5	59.03	0.14
LasR–LBD + E6	57.75	1.43
LasR–LBD + E7	58.12	1.06

## Data Availability

Not applicable.

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
