# Peer review of "Inhibition of Quorum-Sensing Regulator from Pseudomonas aeruginosa Using a Flavone Derivative"

_molecules, 2022, doi:10.3390/molecules27082439_

Round 1
Reviewer 1 Report
The authors prepared a well-written and well-organized manuscript aiming to report a potential chrysin derivative as a potent inhibitor of P. aeruginosa QS system. In the present manuscript, the authors demonstrated, using in vitro and in vivo approaches, that the compound inhibits the binding of the LasR regulator to its associated DNA promoter. In particular, the corresponding inhibitor reported in this study demonstrates better activity in the inhibition of biofilm formation and motility leading to suppression of P. aeruginosa virulence, compared to the previously reported flavonoids.
The manuscript is well-written and well-prepared. I found that the topic of current MS is adequate and acceptable for the journal’s scope. The story is compelling and there is a novelty. It is worthy of publication. However, several major and minor concerns/questions, as mentioned below, need to be addressed by the authors before being considered for publication in this journal.
Major:
First, the authors did not adhere to the format of the journal. Results section shall be presented separately from the discussion section. Please revise the manuscript accordingly to meet the guideline.
Second, as the authors pointed out in their manuscript, the authors used E. coli-based luciferase reporter system to screen for the potential QS inhibitory compounds. While it is common to use the E. coli system for gene expression systems and others in the field of molecular biology, there is almost no explanation on why the authors use their E. coli-based luciferase reporter platform. This shall be discussed properly to provide insights to general readers.
Third, the authors suggested that three xanthone derivatives (D5, D6, D7) and three chrysin derivatives (E5, E6, E7) are potential QS inhibitors after two rounds of screening for further analysis (please see Figure 1c and Figure S2). However, in Figure S2, several compounds also demonstrate the reduced bioluminescence profile, comparatively similar to those six compounds. For example, A12, C5, D1 (this is even better than most of the chosen compounds), D2, D4 (better than D6 and E6). I am curious whether these compounds were also examined or excluded from the further analysis? I could not confirm this in the present manuscript. Please explain this.
Fourth, based on Fig. S3a, the authors proposed that no significant perturbation of bioluminescence was observed for E. coli control strain after incubation with D5-D7 and E5-E7 compounds. However, upon observation of Fig. S3a, I felt disagreed with the authors. It is particularly interesting to see the statistical results of D5 and D7. The authors need to include their statistical analysis on all samples in Fig. S3a and S3b to support their claim.
Next, based on the dose-response results presented in Fig. 4a., the authors decide to use 50 μM compound E6 in the further experiment (Fig. 4b). However, it is interesting to observe the discrepancy of results. While 50 μM of E6 could not significantly suppress the production of LasA protease (Fig 4a), this concentration can downregulate the expression of lasA (Fig.4b). How does this happen? The authors shall provide a compelling explanation about this.
Minor:
Fig. 2. What was in the first lane of the EMSA result? There is a band but no information about what the sample was. Also, what is C30? Please provide information about C30 in the caption. All figures need to stand by themselves without the need to read the content of the manuscript. Lastly, please clarify whether explanations of * and ** are correct or not.
Finally, I recommend that this study shall be considered for publication in this journal after addressing the comments/concerns above.
Reviewer 2 Report
The authors showed extensive and in-depth assays to determine the effect of some synthetic molecules, mainly the one called by them E6, on several virulence factors of Pseudomonas aeruginosa involved in Quorum Sensing (QS). This research is exciting because molecules of this type could help antibiotic therapy against bacteria resistant to the current antibiotics. However, there are still several questions that authors should answer before considering the article for publication on Molecules.
- Sometimes they confuse pathogenesis and virulence, as well as antipathogenic with antivirulence
- The synthesis procedures and spectroscopic properties (NMR and MS) of the molecules assayed as Quorum Sensing regulators are not found.
- There is no explanation for the selection of xanthones since, in the first instance, they were interested in flavonoids, whose high concentration prevents them from being considered as possible antipathogen substances.
- It is also strange that in the case of flavonoids, the starting compounds, such as chrysin, were not synthesized or analyzed as QS, but molecules already derivatized. In addition, the selection of substituents also does not follow a specific pattern to establish possible structure-activity relationships
- There are many reports in the literature about the Quorum Sensing and Quorum Quenching activity of flavonoids and even xanthones; the authors should do a complete review, and especially compare the concentrations used in the assays, since QS is expressed at low concentrations of compounds and 100 mM could be considered as a high concentration
- Finally, molecular docking should be explained more deeply since its analysis and approximation are superficial. Similarly, the interaction shown would not explain the activity of only one derivative. For example, the structural difference between a methyl group and a butyl or ethyl group is minimal and steric.
- According to the figure presented, the amino acid involved in the interaction with the substituent phosphoramidate was not established.
Round 2
Reviewer 1 Report
The authors prepared a well-written and well-organized manuscript aiming to report a potential chrysin derivative as a potent inhibitor of P. aeruginosa QS system. In the present manuscript, the authors carried out an extensive revision on the manuscript and I found that the current form is much better than the previous one. I believe all but one of my comments have been addressed properly. I need confirmation from the authors for below:
My comment in the first round of review: Based on the dose-response results presented in Fig. 4a., the authors decide to use 50 μM compound E6 in the further experiment (Fig. 4b). However, it is interesting to observe the discrepancy of results. While 50 μM of E6 could not significantly suppress the production of LasA protease (Fig 4a), this concentration can downregulate the expression of lasA (Fig.4b). How did this happen? The authors shall provide a compelling explanation about this.
Authors’ Response: This discrepancy between the virulence measurement and gene transcription results is probably due to the different experimental conditions. In virulence measurement, the virulence quantification was performed after 24 h incubation with the inhibitor, while the incubation time is 4 h in the qPCR experiment.
My comment in this second round of review: Why did the authors carry out the experiment at different conditions in the first place? Why not perform the experiment at both time points? I expect this can be done easily.
Reviewer 2 Report
- The synthesis procedures and spectroscopic properties (NMR and MS) of the molecules assayed as Quorum Sensing regulators are not found.
Response:Since all the molecules used in the screening assay have been publish by our group before, the synthesis procedures and characterization are not included in this manuscript. However, we have updated the table S2 in the supporting information to indicate the sources of the molecules.
This explanation must be included in the manuscript in the Experimental Section as: … compounds were obtained using a in house compounds library, and then to add the reference od the paper.
- There is no explanation for the selection of xanthones since, in the first instance, they were interested in flavonoids, whose high concentration prevents them from being considered as possible antipathogen substances.
Response: Previous studies showed that s also exhibited good anti-bacterial activities. And xanthones has a similar structure to flavonoids. Therefore, we also included the xanthones derivatives in the screening. We also added an explanation in line 77 of the revised manuscript.
Ok, but additional QS or QQ effect references of xanthones were forgotten¡
- It is also strange that in the case of flavonoids, the starting compounds, such as chrysin, were not synthesized or analyzed as QS, but molecules already derivatized. In addition, the selection of substituents also does not follow a specific pattern to establish possible structure-activity relationships
Response: Previous studies have showed that chrysin could inhibit the biofilm formation of P. aeruginosa at relatively high concentration, so we started with a self-synthesized library of flavonoids and xanthones derivatives with the hope to improve the anti-QS activity. After picking out the E5, E6 and E7 compounds, we noted that they are all derivatives of chrysin, so that we used chrysin as a control in the biofilm and virulence inhibition assay.
This explanation must be included in the manuscript, because seems to be a bioguided process
Xanthones has a similar structure to flavonoid, so xanthones derivatives were also included the screening.
Is not true. Both compounds displayed a benzochroman moiety, but they are not similar, since a coplanar system is displayed in xanthones, but flavonoid possess a quasi-perpendicular side phenyl ring. This is important in target interactions and in the docking analysis. May be benzochroman moiety is the common pharmacophore
- Finally, molecular docking should be explained more deeply since its analysis and approximation are superficial. Similarly, the interaction shown would not explain the activity of only one derivative. For example, the structural difference between a methyl group and a butyl or ethyl group is minimal and steric. According to the figure presented, the amino acid involved in the interaction with the substituent phosphoramidate was not established.
Response: We further explained the docking results in the revised manuscript (line 214-216, line 256-260). Based on the docking structure, the phosphoramidate group of compound E6 could probably bind to the residue Tyr47, which was located in the loop L3 of LasR and covers the LasR ligand binding pocket. It is implied that compound E6 binding may perturb the conformation of Tyr47 and destabilized the ligand-bound LasR dimer, thereby inducing the dissociation of LasR from bound DNA. We agree with the reviewer’s opinion. It is difficult to explain all the results with a docking structural model that is only based on simulated calculation. The actual complex structure of LasR and the flavonoid warrants further investigation.
Flavonoids possess several phenolic OH to establish H bonding, and aromatic rings to form stacked or pi complex but, according to authors all interaction are hydrophobic. Moreover, Lys and Asp offer other type of interactions. Besides, the authors did not explain specifically the kind of interaction between phosphoramidate group and the Tyr residue. I think that docking analysis is a poor contribution to the paper.
